# Automated Detection and Counting of Wild Boar in Camera Trap Images

**DOI:** 10.3390/ani14101408

**Published:** 2024-05-08

**Authors:** Anne K. Schütz, Helen Louton, Mareike Fischer, Carolina Probst, Jörn M. Gethmann, Franz J. Conraths, Timo Homeier-Bachmann

**Affiliations:** 1Institute of Epidemiology, Friedrich-Loeffler-Institut, Federal Research Institute for Animal Health, Südufer 10, 17493 Greifswald-Insel Riems, Germany; carolina.probst@bmz.bund.de (C.P.); joern.gethmann@fli.de (J.M.G.); franz.conraths@fli.de (F.J.C.); timo.homeier@fli.de (T.H.-B.); 2Animal Health and Animal Welfare, Faculty of Agricultural and Environmental Science, University of Rostock, Justus-von-Liebig-Weg 6, 18059 Rostock, Germany; helen.louton@uni-rostock.de; 3Institute of Mathematics and Computer Science, University of Greifswald, Walther-Rathenau-Straße 47, 17487 Greifswald, Germany; mareike.fischer@uni-greifswald.de; 4Federal Ministry for Economic Cooperation and Development, Stresemannstraße 94, 10963 Bonn, Germany

**Keywords:** computer vision, European wildlife, camera trap, wild boar, animal counting, animal detection

## Abstract

**Simple Summary:**

This study shows that automated computer vision techniques are highly effective when used to analyze images and to extract valuable information from them. We trained an algorithm with a set of 1600 images obtained from a study where wildlife approaching wild boar carcasses were monitored. This enabled the model to detect different classes of animals automatically in their natural environment with a mean average precision of 98.11%.

**Abstract:**

Camera traps are becoming widely used for wildlife monitoring and management. However, manual analysis of the resulting image sets is labor-intensive, time-consuming and costly. This study shows that automated computer vision techniques can be extremely helpful in this regard, as they can rapidly and automatically extract valuable information from the images. Specific training with a set of 1600 images obtained from a study where wild animals approaching wild boar carcasses were monitored enabled the model to detect five different classes of animals automatically in their natural environment with a mean average precision of 98.11%, namely ‘wild boar’, ‘fox’, ‘raccoon dog’, ‘deer’ and ‘bird’. In addition, sequences of images were automatically analyzed and the number of wild boar visits and respective group sizes were determined. This study may help to improve and speed up the monitoring of the potential spread of African swine fever virus in areas where wild boar are affected.

## 1. Introduction

The use of camera traps to monitor and manage wildlife has increased in recent years [1]. It is a cost-effective, non-invasive method for capturing images of wildlife in their natural habitat with little or no disturbance [2,3]. Meanwhile, images and videos from camera traps are widely shared on social media and are also used for citizen science projects [4,5]. Silver et al. [6] used camera traps to estimate jaguar abundance, and Harvey et al. [7] used them for monitoring wild horses. Data from camera traps can also be used to calculate the density of an animal population [8]. Garabedian and Kilgo [9] used camera trap surveys to monitor the recovery of wild boar populations following density reductions. The European Observatory of Wildlife (EOW) estimated wild ungulate densities by camera trapping in 37 European areas [10].

Susceptible host density is a key factor for the transmission dynamics of infectious diseases, including in wildlife. Therefore, knowledge of the density of local populations is crucial for guiding decisions regarding disease prevention and control measures and for assessing their success [11]. In Europe, wild boar are considered important in the epidemiology of African swine fever (ASF). There seems to be a close link between ASF in domestic pig holdings and wild boar abundance and population dynamics [12,13]. In particular, the density of the local wild boar population is considered crucial in the spread of ASF [14] and is used alongside other factors, such as satellite data, for ASF risk assessment [15]. Morelle et al. [16] used camera trap data to study wild boar density and abundance following an ASF outbreak in Poland. Bollen et al. [17] investigated the use of camera traps to monitor wild boar densities in ASF-affected and non-affected areas in Belgium.

Wildlife camera traps can produce large datasets [1]. For instance, in a study of wild horses, over 200,000 camera trap images were manually analyzed by experienced observers [7]. Manual analysis of huge image datasets is expensive, time-consuming and error-prone [18]. An automated evaluation system may help to overcome at least some of the above-mentioned limitations [19,20]. Computer vision techniques have already been demonstrated to yield outstanding results in image analysis for the identification of animal species with high accuracy [20,21].

Norouzzadeh et al. [20] applied computer vision techniques to classify and count animals in the Serengeti (Africa). Their model had a classification accuracy of 93.8% and was trained with the Snapshot Serengeti dataset [22]. Banupriya et al. [23] trained a convolutional neural network (CNN) to identify elephants and cheetahs with accuracies of 79% and 86%, respectively. Both networks identified the animals, but did not locate them in the image. Miao et al. [19] used a CNN to classify 20 African wildlife species with an overall accuracy of 87.5%.

In the present study, we applied computer vision techniques to wildlife images collected by motion sensor camera traps in natural settings. These images were obtained from a study by Probst et al. [24], in which wild boar carcasses were monitored under field conditions. Probst et al. [24] studied the behavior of wild boar toward their dead conspecifics to assess the risk of ASF spreading via wild boar carcasses. We thus focused on typical Central European wildlife species, such as wild boar (*Sus scrofa*), red fox (*Vulpes vulpes*), raccoon dog (*Nyctereutes procyonoides*), deer (i.e., red deer (*Cervus elaphus*), fallow deer (*Dama dama*) and roe deer (*Capreolus capreolus*)) and birds, that visited wild boar carcasses [25]. Open-source databases like the Open Image Dataset V6 [26] or Snapshot Serengeti [22] are not suitable for European wildlife, as only a few European wildlife animals are included and especially not all classes that are considered here. Carl et al. [27] used a pre-trained model (FasteRCNN + InceptionResNetV2 network) for an automated detection of European wild mammal species with a classification accuracy of 71%.

Therefore, the aim of this study was to first train a CNN, YOLOv4 (You only look once version 4) [28], to detect five classes of typical European wildlife species, namely ‘wild boar’, ‘fox’, ‘racoon dog’, ‘deer’ and ‘bird’. Subsequently, the results of the model were used to develop an algorithm to determine the visits and group sizes of wild boar sounders in a sequence of camera trap images.

## 2. Materials and Methods

### 2.1. Class Selection

We defined five classes of animals—‘wild boar’, ‘fox’, ‘racoon dog’, ‘deer’ and ‘bird’—which represent or include the most frequently detected animals. For methodological reasons, we do not use the term “class” in the biological sense of a taxonomic rank, but rather to refer to one species or a group of related species: Due to the limited number of images for some species in the dataset, it was not possible to train the model to identify these taxa. Hence, we combined such species in classes. The classes ‘wild boar’, ‘fox’ and ‘raccoon dog’ include one single species each, namely the wild boar (*Sus scrofa*), red fox (*Vulpes vulpes*) and raccoon dog (*Nyctereutes procyonoides*). The class ‘deer’ includes three species: red deer (*Cervus elaphus*), roe deer (*Capreolus capreolus*) and fallow deer (*Dama dama*). The class ‘bird’ includes a large variety of bird species, e.g., the white-tailed sea eagle (*Haliaeetus albicilla*), raven (*Corvus corax*), buzzard (*Buteo buteo*) and jay (*Garrulus glandarius*).

### 2.2. Image Data

The image set was generated from motion sensor camera traps used in a study to investigate the behavior of wild boar toward their dead conspecifics in the period of October 2015–October 2016 in a limited area in North-Eastern Germany around Greifswald [24]. Most of the cameras were installed on trees at a height of 1 to 2.2 m above the ground, but if this was not possible, they were installed slightly lower. A total of 122,160 images were taken. From these, a subset of 113,716 images was made available for this study. 

The images have a resolution of 3200 × 2400, 4000 × 3000, 2560 × 1920 or 1648 × 1236 pixels (length × height). Five different camera types were used, namely Moultry I40 and Moultry A5 (Moultrie Alabaster, AL, USA), Seissinger Special-Cam 3 Classic (Anton Seissiger GmbH, Würzburg, Germany), Maginon WK 3 HD (Supra, Kaiserslautern, Germany) and Dörr Snapshot UV555 (Dörr GmbH, Neu-Ulm, Germany). Figure 1 illustrates examples of the different image qualities.

From the total set of images, 2000 images were selected. It was ensured that the image set contained at least 200 images per class. These images were taken by different cameras (different resolutions); they displayed different animal classes (namely ‘wild boar’, ‘fox’, ‘raccoon dog’, ‘deer’ and ‘bird’) or showed no animals. The images were sampled from the dataset covering the entire study period and with different light conditions (day and night scenes). The software LabelImg [29] was used for manually labeling animals in these images and assigning them to the classes. To this end, a bounding box (BB) was drawn around each animal in the image and labeled with the animal class. The number of images, the number of labels and the split between the training and test set are listed in Table 1. For this purpose, the images of each animal class were randomly divided into training and test sets in a ratio of 80:20 so that the relative proportions of the different classes in the training and test sets are equal.

The 2000 images have almost a 50:50 ratio of day and night scenes (1011:989 images). This splits the 400 images of the test set into test set day scenes (203 images) and test set night scenes (197 images).

### 2.3. Environment Configuration

The environment configuration used in this study is shown in Table 2. The workflow was developed using a Jupyter notebook [30] and Python 3.6.8 [31].

### 2.4. Model Training and Evaluation of the Model Performance

We used the deep learning algorithm YOLOv4 [28]. YOLO is a one-stage object detection algorithm; i.e., a single convolutional neural network (CNN) is used to process the images, and it is able to directly calculate the localization and classification of the detected objects [32]. Bochkovskiy et al. [28] showed that YOLOv4 is faster and more accurate than other available detectors. The algorithm was trained with the training set and then evaluated with the test set. The parameters of YOLOv4 for training are shown in Table 3.

The trained object detection algorithm returns for each image whether there are detections or not, and if so, the class (‘wild boar’, ‘fox’, ‘racoon dog’, ‘deer’ and ‘bird’), the confidence of each detection and the position of the predicted BB (coordinates of the center and width and height) are given. To verify the performance of the trained wildlife detector, manually labeled images (test set, test set day and test set night) are compared with the automated detection results. This allows the determination of the following values: precision, recall, mean average precision (mAP) and detection speed.

In order to distinguish between true positive and false positive detection, the inter-section over union (*IoU*) was used. To this end, the area of overlap between the predicted BB and the ground-truth BB (manually labeled) is divided by the area of the union of both BBs (see Equation (3)). Whenever *IoU* < 0.5, we assume it to be false positive. Whenever *IoU* ≥ 0.5, we assume it to be true positive. If the model detects nothing in an image, although an object was manually labeled, it is false negative.

The calculations are shown in the following Equations (1)–(6) (for more detail, see [33,34]):(1)$$precision=\frac{TP}{TP+FP}$$
(2)$$recall=\frac{TP}{TP+FN}$$
(3)$$IoU=\frac{area\left({BB}_{p}\right)\cap area({BB}_{gt})}{area\left({BB}_{p}\right)\cup area({BB}_{gt})}$$
(4)$$mAP=\frac{\sum_{c=1}^{C}AP\left(c\right)}{C}$$
where *TP* is the number of true positive detections, *FP* is the number of false positive detections, *FN* is the number of false negative detections, *BBp* is the predicted bounding box (BB) of the model, *BBgt* is the ground-truth BB (i.e., manually labeled), C is the number of classes (here *C* = 5), *AP* is the average precision and *mAP* is the mean average precision. The precision is the ratio of TP to the number of positive predictions (*TP* + *FP*). The recall is the ratio of *TP* to the total number of objects (*TP* + *FN*). The *AP* is the precision average of all recall values between 0 and 1, and can be calculated as the area under the precision–recall curve. Here, we used an 11-point interpolated *AP*:(5)$$AP=\frac{1}{11}\sum_{r\in \left\{0,0.1,\ldots ,1\right\}}p_{interp}\left(r\right)$$
(6)$$p_{interp}\left(r\right)=\underset{\hat{r}:\hat{r}\geq r}{\max}p\left(\hat{r}\right)$$
where *p*($\hat{r}$) is the precision at recall $\hat{r}$.

### 2.5. Automated Wild Boar Counting

For a single image, the trained wild animal detector is applied and the number of detections of the class ‘wild boar’ corresponds to the automatically determined number of wild boar in the image.

For a sequence of images (e.g., camera trap images of a whole day), the detector is applied to each image, and the number of visits of wild boar as well as the group size of wild boar per visit is determined. A visit is defined as any presence of an animal in an image, according to Probst et al. [25]. All images with the same class and a temporal context belong to the same visit: If the time interval from the last detection of the class is longer than eight minutes, the image is assigned to a new visit; otherwise, it is assigned to the same visit. To count the number of wild boar in one visit automatically, the number of wild boar in each image of the visit is determined and ultimately the maximum number is taken.

The workflow of the algorithm for automatic evaluation of the image data and the determination of the number of visits of wild boar with the respective number of wild boar is shown in Figure 2.

## 3. Results

### 3.1. Performance of the Model

The performance of the model was determined by comparing the results of the automatic detection with the manually labeled test set. The performance for each class is shown in Table 4, and the overall performance is shown in Table 5. The *mAP* of the model is 98.11%, the average *IoU* is 0.81, *the precision and recall* are 0.95 and 0.96, respectively, and the detection speed reaches 103 ms per image. Figure 3 shows the detection of each class, i.e., ‘wild boar’, ‘deer’, ‘fox’, ‘bird’ and ‘racoon dog’. The detection works for all classes, single or multiple animals in the same image, only partially visible animals and different lighting conditions. In addition, multiple classes in the same image and wild boar with different fur colors were detected.

The performance of the model for day and night scenes was determined using the day scene or night scene test sets, respectively; the overall performance for both scenes is shown in Table 6. The *mAP* of the model for the day scene test set is 96.62% and for the night scene test set is 98.51%.

### 3.2. Wild Boar Counting

There are several values for counting wild boar that were determined: the number of wild boar per image, the number of wild boar visits and the group size per visit. For a single image, the number of detections was defined as an automated count of wild boar per image. Figure 4 depicts automated detection of wild boar and the number of detections for different images. Figure 4a shows the detection of six piglets, and even partially hidden piglets behind the branches were found. The detection and counting of wild boar in night scene images are shown in Figure 4b,d; in each image, two wild boar are detected—even the ones that are only partially covered in the image. Figure 4c depicts three detected wild boar. Here, the two back wild boar are slightly blurred because they are in motion and the front one has a different fur color. Nevertheless, all three are detected.

#### 3.2.1. Number of Wild Boar Visits

The automated evaluation of an image sequence provides the number of visits as well as the respective group size. In order to analyze the result of the automated evaluation, we evaluated eight image sequences with 9669 images manually and automatically. Thirty-six visits were determined with the manual evaluation. The automatic evaluation identified 34 of these 36 visits. Therefore, the probability of missing a visit is 5.56%. Table A1 in Appendix A presents all determined visits with the respective manual and automated evaluation and the difference in group size.

#### 3.2.2. Number of Wild Boar per Visit

An interesting complementary piece of information is the group size per visit, which is also relevant for the determination of wild boar density. Exemplarily, we considered a visit that was documented with 10 images and a duration of 5 min and 22 s. The automated and manual evaluations for the number of wild boar are shown in Table 7. In addition, two images with BBs of the automatic detection are shown in Figure 5. All images of the visit are attached in Figure A6 in Appendix A.

Regarding the group size, for all 34 detected visits, the difference between automated and manual evaluation is at most 1 (see Table A1). More specifically, for 23 visits the group size is correct. For three visits, one false positive wild boar was detected, and for eight visits one false negative wild boar was detected. During these 34 visits, a total of 121 wild boar were counted manually; in total, the automated evaluation here resulted in 116 counted wild boar. Thus, the total number of wild boar detected automatically is 95.86% of the total number of wild boar counted manually.

## 4. Discussion

This study demonstrates that computer vision techniques can effectively assist in evaluating large image datasets produced by camera trapping under different conditions in natural environments. Our model was trained to detect five classes of European wildlife species, as well as to determine the number of wild boar visits and the number of individual wild boar per visit. The range of species was likely to be influenced by the experimental setting in which the pictures were obtained, i.e., monitoring wild boar carcasses that had been exposed in particular sites in a limited area in North-Eastern Germany [25]. We are confident, however, that the algorithm can also be applied to other datasets if sufficient training data are available for the expected wildlife species.

We trained YOLOv4, an existing open-source object recognition model, to analyze camera trap images. The trained detector has a high mAP for detection and can differentiate between five classes of wildlife animals, namely ‘wild boar’, ‘fox’, ‘racoon dog’, ‘deer’ and ‘bird’, and a detection speed of 103 ms per image. In addition, the detector can classify and locate multiple animals in the same image, as well as different classes in the same image, and animals with different fur colors, such as the wild boar in Figure 3c. Our algorithm for automated evaluation of image sequences demonstrates high sensitivity (i.e., a low probability of missing a visit) and high accuracy (i.e., a low wild boar count error). Furthermore, our approach has the advantage that the evaluation is always the same, in contrast to manual observation [35].

One limitation of counting visits is missed visits, i.e., visit 15 and visit 21 (see Table A1, Appendix A). In both visits, the wild boar are difficult to see even for the human observer, especially in the image of visit 21 (Figure A7) with fog. This limitation could be improved by including images in the training set that show these conditions. Even with better pictures, it is difficult for experienced observers to recognize the exact number of wild boar, and sometimes this is only possible by making comparisons within the visit sequence.

In addition, the AP or mAP can be improved by re-running the training with more images of the classes or in different conditions (e.g., day and night scenes) with a lower AP or mAP. This allows the detector to be adapted to the respective image set.

Due to the limited number of images for some species, it was not possible to train the model for all species identified in the images. We decided to combine species with similar shapes and sizes to test if they can be classified. The study showed that the detection rate for these artificial classes was very high and the model can be used to preselect animal species. If more images are available, the model can also be trained to recognize these species. For wild ruminants grouped as ‘deer’ in this study, species identification was not necessary as these animals represent herbivores. They do not feed on wild boar carcasses and might spread ASFV only mechanically if they come into contact with wild boar carcasses. We assume that the risk of mechanical spread is similar for the members of the ‘deer’ class in the study region. Further differentiation of the class of ‘birds’ is desirable, in particular for taxa feeding on carrion or birds of prey, but was not feasible in this study due to the limited number of images for individual bird species. This limitation may be overcome by increasing the number of images for training or by improving the performance of the algorithm, in this case YOLOv4. 

The determination of the group size of a visit worked accurately. For all determined visits, the group size differs by a maximum of 1 from the manual evaluation (see Table A1, Appendix A). Figure 6b shows a false positive detection (lower left BB) corresponding to an overestimation of the group size. A reason for undetected wild boar may be that some wild boar are overlapping in the images. Figure 6a shows an example of overlapping wild boar; behind the large animal on the left hides another, and there is a small one under its abdomen. These conditions exist in a small number of images, but they are a minor problem in image sequences, where the wild boar are moving and therefore may not be overlapping in every frame. This is supported by the fact that the group size matches in more than half of all visits for the manual and automatic evaluation.

Here, we showed the method of determination of visits and group sizes only for wild boar. The algorithm for determining the group size and the number of visits can be easily adapted to the other animal classes and the length of the time interval between two consecutive detections of the same visit. Here we were particularly interested in wild boar in view of controlling and monitoring ASF. In most cases, hunting statistics are used to draw conclusions about wild boar densities [36]. Therefore, camera traps are a non-invasive method to monitor wild boar. The automated evaluation enables a very fast evaluation of camera trap images. The detection speed of 103 ms per image makes it possible to evaluate large image data quickly; e.g., it takes less than three hours to evaluate 100,000 images. Automatically analyzed data can be used directly to calculate wildlife densities, for example, using the technique of Rowcliffe et al. [8] for calculating animal densities based on camera trap rates. Determination of wild boar density in an area may be part of a follow-up study.

For the determination of the group size, no distinction is made between individuals, and furthermore, no distinction is made between piglets and adult wild boar or males and females. It is possible to determine the respective number of piglets and adults per visit and not only the total group size with the manual evaluation. In further studies, a detector could be trained to distinguish between piglets and adults or even individual animals.

## 5. Conclusions

This study shows that automated computer vision techniques can support rapid and effective analysis of large image datasets generated by camera traps with respect to Central European wildlife. The method is suitable for many different research questions. Here we showed that the method was able to detect different classes as well as the number of each class. Regarding wild boar, the technique provides the possibility of determining the number of wild boar visits and the respective group sizes in an automated way. The trained model achieves a high mAP. The detection speed of 103 ms per image makes it possible to evaluate 9669 images (see Section 3.2. Wild Boar Counting) in less than 20 min, compared to several days evaluating the images manually. The automatic evaluation of the number of wild boar visits and the group sizes provides accurate results.

These results may be used in the field of risk analysis, animal disease control and surveillance to estimate animal densities rapidly and reliably. The trained detector achieves a mean average precision of 98.11%. The results of detection can be used to count the number of animals per class per image. In addition, the number of visits and group size can be analyzed automatically. This study may help to improve and speed up the monitoring of the potential spread of ASF in areas where wild boar are affected by the disease.

## Figures and Tables

**Figure 1 animals-14-01408-f001:**
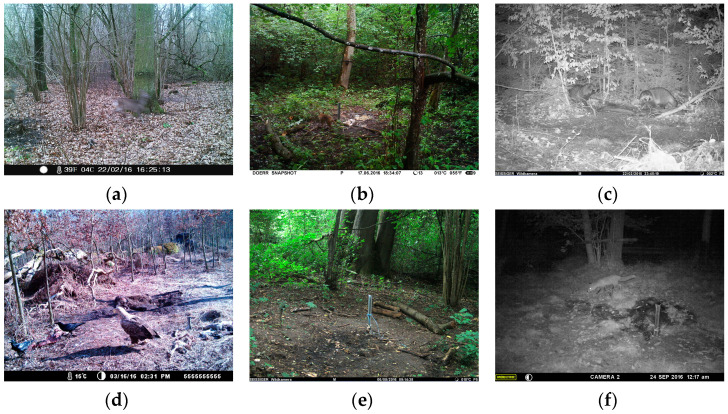
Examples of camera trap images, diverse camera types, different seasons, day scenes (**a**,**b**,**d**,**e**) and night scenes (**c**,**f**). The camera types used were Maginon WK3 HD (**a**), Dörr Snapshot UV555 (**b**), Seissinger Special-Cam 3 Classic (**c**,**e**), Moultry I40 (**d**) and Moultry A5 (**f**). For details, please refer to [24].

**Figure 2 animals-14-01408-f002:**
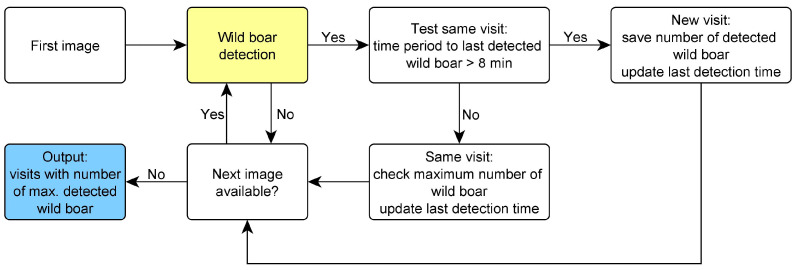
Workflow image data evaluation. Determination of the number of visits of wild boar and the number of wild boar per visit. One image after the other is analyzed with the detector until all images are evaluated. If wild boar detection is positive, it is tested if the last wild boar detection was less than 8 min ago, in which case it is the same visit. Otherwise, it is a new visit. For a visit that has already started, the maximum number of wild boar in the visit is compared with the number in the current image and updated if necessary. For a new visit, the number of detected wild boar is saved as the maximum number of wild boar. In both cases, the last detection time is updated.

**Figure 3 animals-14-01408-f003:**
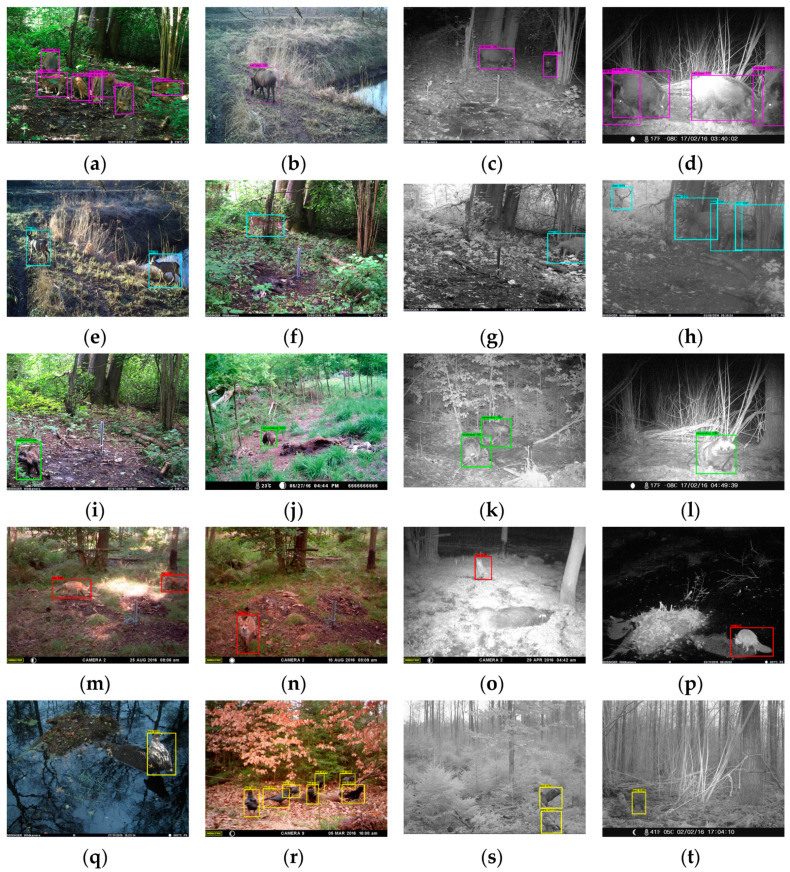
Detection examples for each class, two day scene images and two night scene images each. The BBs are depicted as colored boxes. ‘Wild boar’: (**a**–**d**), ‘deer’: (**e**–**h**), ‘raccoon dog’: (**i**–**l**), ‘fox’: (**m**–**p**) and ‘bird’: (**q**–**t**). (All images are shown in higher resolution in Appendix A, attached in Figure A1, Figure A2, Figure A3, Figure A4 and Figure A5).

**Figure 4 animals-14-01408-f004:**
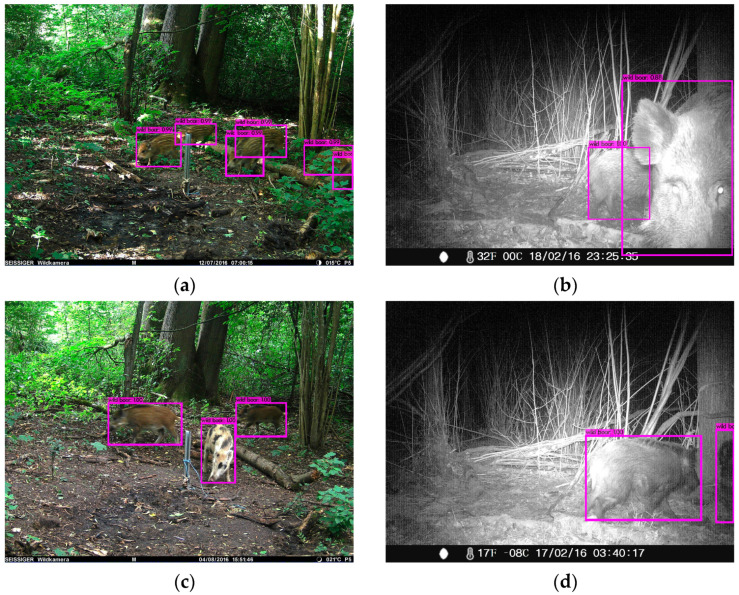
Example of automated wild boar detection and group size estimation in complex settings. The BB depicted as a purple box. (**a**) the detection of six piglets, (**b**,**d**) detection of wild boar in night scene images; in each image, two wild boar are detected, (**c**) three detected wild boar, one with a special fur color.

**Figure 5 animals-14-01408-f005:**
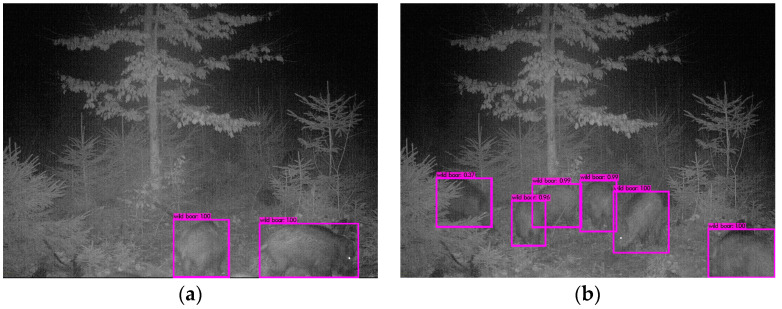
A visit of wild boar with a group of 6 animals. (**a**) First image of the visit. Two wild boar were automatically detected. (**b**) Image with the largest number of wild boar during the visit; 6 wild boar were detected automatically. The BB depicted as a purple box.

**Figure 6 animals-14-01408-f006:**
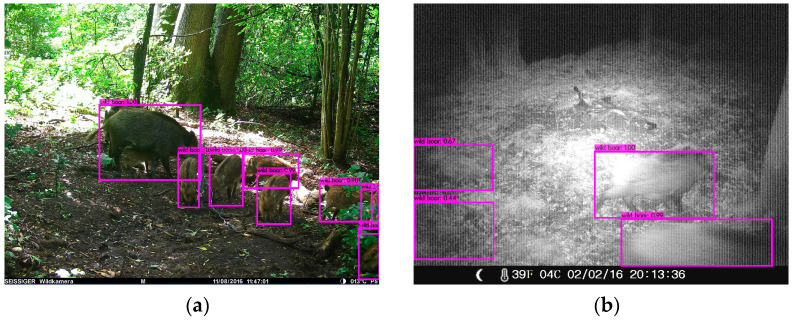
(**a**) Missed detection—overlapping wild boar; (**b**) additional detection. The BB depicted as a purple box.

**Table 1 animals-14-01408-t001:** Splitting of image set (2000 images with labels) into training and test sets in the ratio 80% to 20%. The numbers of images and labels per animal species and the numbers of images without animals for the training set and test set are also shown. The relative proportion of images and labels for each class shows that the distribution of classes is almost the same in both datasets. Note: In both image sets, there are images with several animal species; e.g., in the test set, there is an image showing both a raccoon dog and a fox.

	Training Set		Test Set	
	Number of Images (Relative Proportion in %)	Number of Labels (Relative Proportion in %)	Number of Images (Relative Proportion in %)	Number of Labels (Relative Proportion in %)
Wild boar	522 (32.6)	1330 (54)	130 (32.5)	318 (53.1)
Fox	474 (29.6)	481 (19.5)	119 (29.8)	119 (19.9)
Raccoon dog	197 (12.3)	263 (10.7)	50 (12.5)	65 (10.9)
Deer	160 (10)	176 (7.1)	40 (10)	45 (7.5)
Bird	160 (10)	215 (8.7)	40 (10)	52 (8.7)
No animal	88 (5.5)	0 (0)	22 (5.5)	0 (0)
Total	1600	2465	400	599

**Table 2 animals-14-01408-t002:** Environment configuration used in this study for model training and automatic evaluation of the images.

	Specification
Operating system	CentOS 7
Graphics card	NVIDIA K80 with 2 GPUs and 24 GB video RAM
Processor	Intel Xeon E5-2667 v4 with 3.20 GHz
RAM	377 GB

**Table 3 animals-14-01408-t003:** Parameters of YOLOv4 for the wild animal detection model.

Parameter	Value
Input size	416 pixels × 416 pixels
Classes	5
Maxbatches	10,000
Filters	30
Steps	8000 and 9000
Learning rate	0.001
Batch size	64

**Table 4 animals-14-01408-t004:** Performance of the trained model for the distinct classes ‘wild boar’, ‘fox’, ‘racoon dog’, ‘deer’ and ‘bird’ with *AP*—average precision and *IoU*—intersection over union.

Class	*Precision*	*Recall*	*Average IoU*	*AP*
Wild boar	0.95	0.94	0.82	96.41%
Fox	0.95	1.0	0.83	99.88%
Racoon dog	0.92	0.97	0.74	96.53%
Deer	0.98	0.91	0.84	98.08%
Bird	1.0	0.98	0.83	99.64%

**Table 5 animals-14-01408-t005:** Overall performance of the trained model (with *IoU*—intersection over union and *mAP*—mean average precision).

*Precision*	*Recall*	*Average IoU*	*mAP*	Detection Speed
0.95	0.96	0.82	98.11%	103 ms

**Table 6 animals-14-01408-t006:** Overall performance of the trained model for day and night scenes (with *IoU*—intersection over union and *mAP*—mean average precision).

	*Precision*	*Recall*	*Average IoU*	*mAP*
Day	0.98	0.95	0.85	96.62%
Night	0.95	0.97	0.82	98.51%

**Table 7 animals-14-01408-t007:** Manual and automatic counting of wild boar for all images of a visit (corresponds to visit 9 from Table A1). The maximum group sizes of the visit for both of the two counting methods are shown in bold.

Image	Automatic Count	Manual Count	Difference
1	2	2	0
2	4	4	0
3	4	4	0
4	3	4	1
5	**6**	**6**	0
6	5	5	0
7	3	4	1
8	4	4	0
9	3	4	0
10	1	3	2

## Data Availability

The datasets found or analyzed during the current study are available from the corresponding author on reasonable request.

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
