# Peer review of "Automated Detection and Counting of Wild Boar in Camera Trap Images"

_animals, 2024, doi:10.3390/ani14101408_

Round 1
Reviewer 1 Report
Comments and Suggestions for Authors
I found the paper very interesting and informative. It was very well written; however, I have some issues, which I will detail below.
Line 63: OK – this is the first time you mention accuracy. This would be a good time to define the key terms. You use precision and recall – which one of these is accuracy? Do they work it out the same way?
After working through the equations, I understand your terms more, but I think you should explain all the key terms in more detail here. Also, I kept thinking about which one is most important. I guess it depends on the situation. For example, if you are trying to find a reinvading rat on an offshore island, you would favour recall because you don’t want any false negatives. Which one do you think is most important in detecting and counting wild boar?
Line 107: You mention five different camera types. I have found big differences between trail camera brands. While you have balanced the test set for animal class and day/night, Could you also do this for the camera model and explicitly test the effect of the camera type? I have also found that the distance the animal is from the camera can influence ID. Do you think different camera sets could have influenced the results? Also, were the camera settings consistent with the number of images, delays between photos, etc.?
Line 159: I gather that you use IoU to find false positives. Do you ground truth this – were values less than 0.5 always a false positive?
Line 173: I understand that researchers use time intervals to determine the number of visits. These intervals tend to vary between species – has this been tested for pigs? Would changing the interval to 10 minutes or longer make any difference? Should you test this?
Line 238: Reference missing here.
Line 311: I agree that cameras are non-invasive. But I think you should talk about camera impact and baited sites. I have just finished a study where I had many animals interacting with the cameras – I am sure they can hear or see them. Do you think the placement of cameras could influence natural behaviour, or is this not the case for boar, given they are probably keen to visit the site?
Reviewer 2 Report
Comments and Suggestions for Authors
It should be noted that the authors undertook quite interesting research that may have both application and practical importance, especially since it concerns the increasingly widely used research methods using camera traps for environmental research. The added value of the manuscript submitted for evaluation is the fact that this research concerns the wild boar population, which is important when undertaking all kinds of activities related to detecting, limiting the spread and possible combating of African swine fever. Due to the very high adaptability of this species to its habitats, all research and information on this subject is valuable and allows for supplementing existing knowledge in this area.
After reading the manuscript, I have some observations and comments:
There is no mention of ASFV in the title, but it is quite strongly emphasized in the summary, text and literature. Additionally, the application also mentions the use of this method for disease monitoring. Should you therefore consider changing the title to include these aspects or rewording the study to omit any disease issues?
The methodology does not explain very well the issues related to:
Ø with the ability to detect individual species and their number in a given event and to join and disconnect animals from the area covered by photography (movement of animals within a group - one event),
Ø with the ability to detect animals of different species being in different numbers within the range of a given camera at the same time,
Ø it is not understandable why the authors assumed 8 minutes for one visit of the same individual (where does this come from - maybe there is some literature on this subject?), especially when it comes to birds (e.g. the jay mentioned above can come several or a dozen times in 8 minutes to a given place), so these results will be distorted, especially when there are wild boars or other animals there at the same time, e.g.
Ø what about animals located in the photos deep in the image, can they be recognized by the system? Sometimes we have problems visually recognizing such animals, especially at night, and as we know, animal activity occurs mainly at night.
The description provided by the authors does not convince me of such high precision, especially when photographing animals in various positions and lighting. What about rain, fog and other phenomena that limit recognition?
The authors state that they hung the cameras in trees at a height of 1-2.2 meters, and photo 4B is certainly taken from a lower height, so how were these cameras really placed?
Where were the dead wild boars taken from and on what principles were they disposed of?
After all, they were carrion, didn't it pose an additional epidemiological threat? Was ASF present in the study area? How much time did these dead wild boars or their remains spend in the field, and what was done with the remains?
We only find out where the experiment was conducted in the discussion, but the detailed location as well as the lack of description of the environment where the cameras were hung? After all, wild boars and other species use different natural environments with different pressure and activity.
Is there any information about other animals (species) that certainly appeared with the carcass, including domestic animals (dogs, cats)? Hence, it is worth indicating the places of the experiment.
How long had these dead remains of wild boars been lying there, and why were the dead remains supposed to attract other animals? What was the assumption, was this method only tested to assess wild boars or other animals (no explanation of this key issue).
The discussion chapter does not meet the standards of this type of part of the study. In this chapter, the authors provide methodological and resultant data, and only four times quote studies by other authors. They do not even try to discuss the obtained results with others in this field and only then draw a conclusion, which is generally known and concerns camera traps as extensive cognitive material for the assessment of a given population of wild animals. In my opinion, the only innovation is the proposed computer model, which seems to be omitted in the discussion.
The literature is somehow dominated by studies on ASFV, and the study, as I have already mentioned, does not directly address it, so it seems necessary to supplement it with studies related to the topic of the work.
After reading the entire material, I am concerned about the detection rate of 98.11% stated by the authors. My anxiety results from the so-called false detections, or to make sure it was properly verified, I indicated the possibility of no detection or a mistake earlier and this needs to be explained.
Reviewer 3 Report
Comments and Suggestions for Authors
Although this research doesn't add significant novelties to the topic, it reports useful findings to everyone in the field who want to improve and plan its camera trapping study and enhance the challenging process of image evaluation. The manuscript is rather technical so its crucial to meticulously introduce all necessary details. I gladly noted that this task was well done here. The data specification, the used camera settings and parameter tuning was clearly reported everywhere, just like the performance metrics.
I wonder whether the European badger (Meles meles) was missing from the area where the images were taken?
Since the Authors also stated that open-source databases have very low amount of references about characteristic European wildlife species, it would be advantageous to create a repository to fill the void in the future.
Analyzing static images (photos like these) can be challenging even for a well-trained algorithm or an expert when the bodies of the individuals from the same species (e.g. piglets) are totally or mostly overlap, or obscured by vegetation.
Is there any result whether short videos instead of static images enhance the performance of these algorithms?
Specific comments:
Figure 1: Maybe indicate which image belongs to which camera type to make it even more informative and help choosing the ideal camera for different purposes.
Table 1: Recommend to report the relative proportion of images and labels for each class both for the training and test set to emphasize that the distribution of classes were almost the same in both dataset.
Line 172-173: "If the time interval to the last detection of the class was longer than eight minutes, the image was assigned to a new visit, otherwise..."
Due to this rule of thumb approach (8mins) the reliability of wild boar density estimations can be low based only on this method.
E.g. if one individual from a herd of 6 wild boars leave the detection area, but another one come in range what was never seen before in 8 mins after that first member left, then it will still count as a same detection.
This and similar nuances can add some bias for the evaluation.
Can you provide some data about how large this bias can be?
line 197-198: "In addition, multiple classes in the same image and wild boar with different fur color were detected."
Was it challenging to the model to make distinction in these cases? (Is there any precision or IoU data available or these occasions were rare?)
Table 4, line 6: How can the precision value for 'Bird' reach a higher value than 1?
Table 6:
What caused the contrasting tendency in precision vs. mAP between day and night (precision was lower at night contrary to mAP)? - Less species (classes) were detected at night I suppose.
Line 238-239: 'Error! Reference source not found.' should be Table 7
Table 7:
The last line in the table (maximum) is not necessary, it only repeats information from line 6 (image 5). Recommend to use other solutions (e.g. bold text and explanation in the captions).
Line 336-341: Correct the font size.
Round 2
Reviewer 2 Report
Comments and Suggestions for Authors
Despite the changes and corrections, I still have serious comments about the manuscript:
1. There is no mention of ASFV in the title, and it is quite strongly emphasized in the summary, text and literature, so in my opinion it is absolutely necessary to change the title and purpose of the work?
2. The authors still have not clarified the issues I raised in the previous review, namely:
The methodology does not explain very well the issues related to:
· with the ability to detect individual species and their number in a given event and to join and disconnect animals from the area covered by photography (movement of animals within a group - one event),
· with the ability to detect animals of different species being in different numbers within the range of a given camera at the same time,
· it is not understandable why the authors assumed 8 minutes for one visit of the same individual (where does this come from - maybe there is some literature on this subject?), especially when it comes to birds (e.g. the jay mentioned above can come several or a dozen times in 8 minutes to a given place), so these results will be distorted, especially if there are wild boars or other animals there at the same time, e.g.
· what about animals located in the photos deep in the image, can they be recognized by the system? Sometimes we have problems visually recognizing such animals, especially at night, and as we know, animal activity is mainly at night.
Please respond to this, at least in the form of explanations and a mention in the text.
3. The following issues were also not clarified:
Where were the dead wild boars taken from and on what principles were they disposed of? After all, they were carrion, didn't it pose an additional epidemiological threat? Was ASF present in the study area? How much time did these dead wild boars or their remains spend in the field, and what was done with the remains?
4. The discussion chapter does not meet the standards of this type of part of the study. For example, Line 286-319, these are typical statements, not a discussion. So why include it in the discussion, maybe it's better to add it to the results or leave it out altogether?
Due to the fact that the authors did not explain the issues raised in the previous review and did not improve the work according to the suggestions provided, I propose that the work be rejected.
Author Response
Dear Reviewer,
Thank you for your comments.
- The study is about automatically determining the number of visits and the respective group size. We are therefore of the opinion that the ASF should not be included in the title.
- In our opinion, these points are sufficiently clarified in the manuscript
- This information is part of the study from which the images originate, please see Probst et al. 2017.
- As the other reviewers had no objections to the discussion, we would like to leave it at that